# High throughput ANI analysis of 90K prokaryotic genomes reveals clear species boundaries

Chirag Jain[1,2], Luis M. Rodriguez-R [3,4], Adam M. Phillippy[2], Konstantinos T. Konstantinidis[3,4] & Srinivas Aluru[1,5]

A fundamental question in microbiology is whether there is continuum of genetic diversity among genomes, or clear species boundaries prevail instead. Whole-genome similarity metrics such as Average Nucleotide Identity (ANI) help address this question by facilitating high resolution taxonomic analysis of thousands of genomes from diverse phylogenetic lineages. To scale to available genomes and beyond, we present FastANI, a new method to estimate ANI using alignment-free approximate sequence mapping. FastANI is accurate for both finished and draft genomes, and is up to three orders of magnitude faster compared to alignment-based approaches. We leverage FastANI to compute pairwise ANI values among all prokaryotic genomes available in the NCBI database. Our results reveal clear genetic discontinuity, with 99.8% of the total 8 billion genome pairs analyzed conforming to >95% intra-species and <83% inter-species ANI values. This discontinuity is manifested with or without the most frequently sequenced species, and is robust to historic additions in the genome databases.

[1] School of Computational Science and Engineering, Georgia Institute of Technology, Atlanta, GA 30332, USA. [2] National Human Genome Research Institute, National Institutes of Health, Bethesda, MD 20894, USA. [3] School of Civil and Environmental Engineering, Georgia Institute of Technology, Atlanta, GA 30332, USA. [4] School of Biological Sciences, Georgia Institute of Technology, Atlanta, GA 30332, USA. [5] Institute for Data Engineering and Science, Georgia Institute of Technology, Atlanta, GA 30332, USA. Correspondence and requests for materials should be addressed to K.T.K. (email: kostas.konstantinidis@gatech.edu) or to S.A. (email: aluru@cc.gatech.edu)

Large collections of prokaryotic genomes with varied ecologic and evolutionary histories are now publicly available. This deluge of genomic data provides the opportunity to more robustly evaluate important questions in microbial ecology and evolution, as well as underscores the need to advance existing bioinformatics approaches for the analysis of such big genomic data. One such question is whether bacteria (and other microbes) form discrete clusters (species), or, due to high frequency of horizontal gene transfer (HGT) and slow decay kinetics, a continuum of genetic diversity is observed instead. Studies based on a small number of closely related genomes have shown that genetic continuum may prevail[1]. On the other hand, other studies have argued that HGT may not be frequent enough to distort species boundaries, or that organisms within species exchange DNA more frequently compared to organisms across species, thus maintaining distinct clusters[2]. An important criticism of all these studies is that they have typically been performed with isolated genomes in the laboratory that may not adequately represent natural diversity due to cultivation biases, or were based on a small number of available genomes from a few phylogenetic lineages, which does not allow for robust conclusions to emerge. Therefore, it is still unclear if well-defined clusters of genomes are evident among prokaryotes and how to recognize them. Defining species is not only an important academic exercise but also has major practical consequences. For instance, the diagnosis of disease agents, the regulation of which organisms can be transported across countries and which organisms should be under quarantine, or the communication about which organisms or mixtures of organisms are beneficial to human, animals, or plants, are all deeply-rooted on how species are defined.

One fundamental task in assessing species boundaries is the estimation of the genetic relatedness between two genomes. In recent years, the whole-genome average nucleotide identity (ANI) has emerged as a robust method for this task, with organisms belonging to the same species typically showing ≥95% ANI among themselves[3,4]. ANI represents the average nucleotide identity of all orthologous genes shared between any two genomes and offers robust resolution between strains of the same or closely related species (i.e., showing 80–100% ANI). The ANI measure does not strictly represent core genome evolutionary relatedness, as orthologous genes can vary widely between pairs of genomes compared. Nevertheless, it closely reflects the traditional microbiological concept of DNA–DNA hybridization relatedness for defining species[3], as it takes into account the fluid nature of the bacterial gene pool and hence implicitly considers shared function.

Sequencing of 16S rRNA genes is another highly popular, alternative traditional method for defining species and assessing their evolutionary uniqueness. However, methods based on single[5] or a set of universally conserved genes[6], such as 16S rRNA and ribosomal protein-encoding genes are often not applicable to incomplete genomes (e.g., the genes are not assembled), and these genes typically show higher sequence conservation than the genome average. Consequently, analysis of universal genes does not provide sufficient resolution at the species level[7], and has frequently resulted in lack of clear genetic discontinuities among closely related taxa[6]. ANI offers several important advantages such as higher resolution among closely related genomes. Finally, ANI can be estimated among draft (incomplete) genome sequences recovered from the environment using metagenomic or singe-cell techniques that do not encode universally conserved genes but encode at least a few hundred shared genes, greatly expanding the number of sequences that can be studied and classified compared to a universal gene-based approach. Accordingly, ANI has been recognized internationally for its potential for replacing DNA–DNA hybridization as the standard

measure of relatedness, as it is easier to estimate and represents portable and reproducible data[8,9]. Despite these strengths, to date ANI-based methods could not be applied for a large number of genomes due to their reliance on alignment-based searches [e.g., BLAST[10]], which are computationally expensive due to the quadratic time complexity of alignment algorithms[11]. As such, faster alternatives to BLAST such as BLAT[12], Usearch[13] or DIAMOND[14] (after translation to protein level) for computing nucleotide alignments also suffer from the same limitation.

Several variations of the original ANI calculation algorithm have been proposed[15–18], however these mainly modify the specific approach to identify shared genes and do not speedup the calculation substantially since they are all alignment-based. Accordingly, it is nearly impossible to calculate ANI values among the available microbial genomes to date, in the order of a hundred thousand, based on these approaches and commonly available computational resources. Importantly, the available genomic data is estimated to be a small fraction of the extant prokaryotic diversity[19,20], and the number of new genomes determined continues to grow exponentially. Therefore, new computational solutions are needed to scale-up and comprehensively analyze the available and forthcoming data.

A couple of such solutions have been proposed recently, borrowing concepts from 'big data' analysis in other scientific domains. MinHash is a technique for quick estimation of similarity of two sets, initially developed for the detection of near-duplicate web documents in search engines at the scale of the World Wide Web[21]. Recently, this technique was successfully adapted for designing new fast algorithms in bioinformatics such as for genome assembly[22,23] and long read mapping problems[24]. Ondov et al.[25] provided the first proof-of-concept implementation called Mash for fast estimation of ANI using this technique. Even though Mash has been reported to be multiple orders of magnitude faster than alignment-based ANI computation, a straight-forward adoption of the MinHash technique to the problem of computing ANI has been found to be inaccurate for incomplete draft genomes[26]. Further, there is a limit on how well Mash can approximate ANI especially for moderately divergent genomes (e.g., showing 80–90% ANI), as Mash similarity measurement is not restricted to the shared genomic regions, whereas ANI considers only the shared genome.

In this study, we alleviate the computational bottleneck in ANI computation by developing FastANI, a novel algorithm utilizing Mashmap[24] as its MinHash based alignment-free sequence mapping engine. FastANI provides ANI values that are essentially identical to the alignment-based ANI values for both complete and draft quality genomes that are related in the 80 to 100% nucleotide identity range. Therefore, FastANI enables accurate estimation of pairwise ANI values for large cohorts of genomes or evaluation of the novelty of a query draft genome by comparing it against the full collection of available prokaryotic genomes.

## Results

**Datasets.** To test accuracy and speed, we evaluated FastANI on both high-quality closed genomes from NCBI RefSeq database as well as publicly available draft genome assemblies. We first removed poor quality genome assemblies with low N50 length (<10Kbp). In total, five datasets were used, D1 through D5 (see Table 1). Dataset D1 is the set of closed prokaryotic genomes downloaded from RefSeq database. Datasets D2, D3, and D4 include draft genome assemblies of isolates of *Bacillus cereus s.l.*, *Escherichia coli*, and *Bacillus anthracis*, respectively, downloaded from the prokaryote section of the NCBI Genome database. Dataset D5 includes a recently published large collection of metagenome-assembled genomes (MAGs)[27]. These sizable

**Table 1 Datasets used for testing accuracy and speed of FastANI**

| Id | Reference clade | No. of genomes | Median N50 (Mbp) | Query genome |
|---|---|---|---|---|
| D1 | NCBI RefSeq | 1675 | 3.14 | *E. coli* K-12 MG1655 |
| D2 | *Bacillus cereus s.l.* | 570 | 1.16 | *B. anthracis* 52-G |
| D3 | *Escherichia coli* | 4271 | 0.15 | *E. coli* 0.1288 |
| D4 | *Bacillus anthracis* | 464 | 0.59 | *B. anthracis* 2000031001 |
| D5 | MAGs[27] | 7897 | 0.04 | *Acinetobacter* sp. UBA6007 |

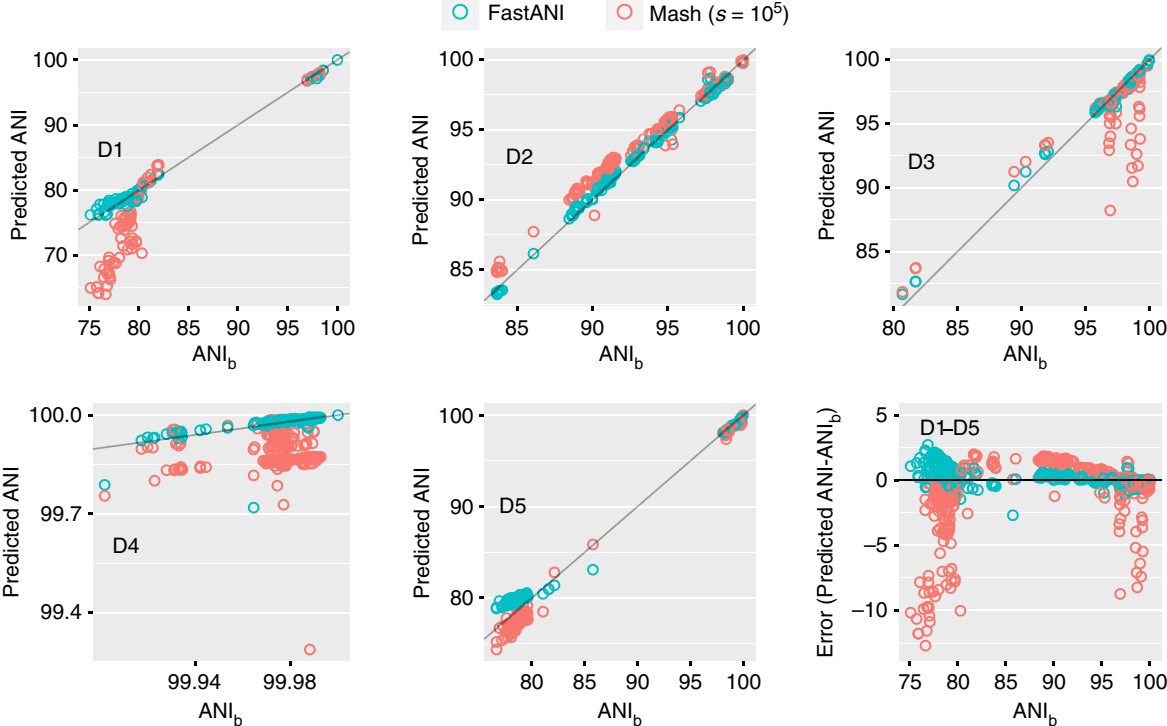

**Fig. 1** Correlation of FastANI and Mash-based ANI output with $ANI_b$ values for datasets D1–D5. Because FastANI assumes a probabilistic identity cutoff that is set to 80% by default, it reports 76, 570, 4271, 464, and 130 genome matches for the individual queries in datasets D1–D5 respectively. To enable a direct quality comparison against FastANI, Mash is executed for only those pairs that are reported by FastANI. Notice that each dataset encompasses a different nucleotide identity range (x-axes). Gray line represents a straight line $y = x$ plot for reference. Pearson correlation coefficients corresponding to these plots are listed separately in Table 2. Last plot shows error of these methods *w.r.t.* $ANI_b$ using all five datasets

datasets represent genomes showing different levels of identity among themselves and varying values of completeness and assembly quality (Supplementary Fig. 1). For each dataset, one genome was selected as the query genome and its ANI was computed with every genome in the complete dataset. In all cases except in D1, query genome strains were selected randomly.

**Accuracy**. We evaluated FastANI against the BLASTn based method[16] of computing ANI, henceforth referred as $ANI_b$, and the ANI values predicted by the Mash[25] (v1.1) tool. User documentation for Mash recommends using larger sketch size (i.e., k-mer sample) than the default to obtain higher accuracy[25]. Accordingly, we ran Mash with both the default sketch size of 1 K as well as increase it up to 100 K.

FastANI achieves near perfect linear correlation with $ANI_b$ on all datasets D1–D5 (Fig. 1 and Table 2). Mash results improve with increasing sketch size, particularly for D1. However, even when executed with the largest sketch size of 100 K, Mash results diverge from $ANI_b$ values on datasets D1, D3, and D4. For D1, this primarily appears to be caused by divergent genomes (e.g., showing <90% ANI). For D3, Mash diverges on closely related

genomes due to fragmented and incomplete genome assemblies of the draft genomes. Dataset D4 is challenging because its constituent genomes are closely related strains of *Bacillus anthracis*, with $ANI_b$ > 99.9 for all the pairs. FastANI provides much better precision than Mash in D4 dataset, and therefore, can be used to discriminate between very closely related microbial strains such as those of different epidemic outbreaks. However, for two genomes out of the 464, FastANI estimates are diverging from $ANI_b$. To investigate further, we visualized gene synteny pattern using Mauve[28] and found that these two genome sequences have many re-arrangements with respect to the query genome (Supplementary Fig. 2). Given that *B. anthracis* strains typically show high genome synteny[29], these results indicate that the two genomes were poorly assembled. Incorrect data will yield unpredictable results not only with FastANI but using any method that assesses genetic relatedness, including phylogeny-based methods. If the two incorrect *B. anthracis* assemblies are removed, FastANI's correlation with $ANI_b$ improves to 0.944 in D4.

These correlation results demonstrate that FastANI provides significant quality improvement over Mash (see Table 2), and can

be a reasonable substitute for $ANI_b$. Although this experiment was conducted with single query genome per dataset, increasing the count of query genomes did not affect our conclusions (Supplementary Fig. 3). Further, FastANI estimates were accurate for draft genomes, in the range of 20–100% completeness based on real (Supplementary Fig. 4) or simulated datasets (Supplementary Fig. 5), implying that FastANI can tolerate variable assembly quality, completeness, and contamination. Most importantly, it correlates well with $ANI_b$ in the desired identity range of 80–100%.

**Computational speedup.** FastANI is designed to efficiently process large assembly datasets with modest compute resources. For FastANI's sequential and parallel runtime evaluation, we used a single compute node with two Intel Xeon E5-2698 v4 20-core processors. First, we show runtime comparison of FastANI and $ANI_b$ using serial execution (single thread, single process) using all datasets in Table 3. FastANI operation consists of indexing phase followed by compute phase, for which we measured the runtime separately. For any database, indexing all the reference genomes needs to be done only once, and thereafter, FastANI can compute ANI estimates for any number of input query genomes against the reference genomes. Therefore, speedup in Table 3 is measured with respect to FastANI compute time. We observe that the runtime improvement due to FastANI varied from 50x for D3 to 4608x for D5. FastANI speedup is much higher on D1 and D5 because these datasets contain a diverse set of prokaryotic genomes. This is attributable to the fact that the algorithm underlying FastANI is able to prune distant genomes $(ANI \ll 80\%)$ efficiently. On the contrary, ANI values for all genomes in datasets D2-D4 were high (>80%). Note that replacing BLASTn with faster alignment software in $ANI_b$ does not improve its performance significantly. A recent survey of ANI methods[18] reported speedups of only up to 4.7x by using Usearch[13] and MUMmer[30],

which is also accompanied with lower accuracy among moderately related genomes in the 75–90% ANI range.

To accelerate ANI computation even further, FastANI can be trivially parallelized using multi-core parallel execution. One way to achieve this is to split the reference genomes in several equal-size parts. This way, each instance of FastANI process can search query genome(s) against each part of the reference database independently. We utilized this scheme and evaluated scalability using up to 80 FastANI parallel processes. Compared to the sequential execution time listed in Table 3, runtime of the compute phase reduced to 2, 8, 46, 6, and 1 s for datasets D1–D5, respectively (Fig. 2). These results confirm that FastANI can be used to query against databases containing thousands of genomes in a few seconds.

For the above experiments, FastANI required a maximum 62 GB memory for D5, our largest dataset for this experiment. For databases much larger than D5, peak memory usage can be reduced by either distributing the compute across multiple nodes in a cluster or processing chunks of the reference database one by one, as necessary.

**Large-scale pairwise comparison indicates genetic discontinuity.** We examined the distribution of pairwise ANI values between all 91,761 prokaryotic assemblies that existed in the NCBI Genome database as of 15 March 2017. Prior to analysis, we removed 2262 genomes due to short N50 length (<10 Kbp). In our evaluation, the ANI between each pair of genomes A and B is computed twice, once with $A$ as query genome and again with $B$ as query genome. This choice did not meaningfully alter the ANI value reported by FastANI unless the draft genomes are incorrectly assembled or contaminated (Supplementary Fig. 6). Computing pairwise ANI values for the entire database took 77 K CPU hours for all 8.01 billion comparisons. To our knowledge, this is the largest cohort of genomes for which ANI has been computed. In comparison, the largest previously published ANI analysis included 86 million comparisons and took 190 K CPU hours[7]. Among the total of 8.01 billion pairwise comparisons, 679,765,100 yielded ANI values in the 76–100% range. The distribution of these ANI values reveals a discontinuity, i.e., the resulting ANI values show a strong bimodal distribution, with a wide gap or lack of values between the two peaks of the distribution. Specifically, FastANI reported only 17,132,536 ANI values (i.e., 2.5% of the 679,765,100 pairs) within the range of 83 to 95%. When performing this analysis using Mash, the bimodal distribution of ANI values was persistent (Supplementary Fig. 7).

The frequency of intra- vs. inter-species genomes sequenced in the NCBI database has changed over time, with earlier sequencing efforts targeting distantly related organisms in order to cover phylogenetic diversity while efforts in more recent years targeted more closely related organisms for micro-diversity or epidemiological studies (Supplementary Fig. 8). We confirmed that discontinuity pattern has been maintained with genome sets from different time points in the last ten years (Fig. 3b). In previous taxonomic studies, 95% ANI cutoff is the most frequently used standard for species demarcation. Density curves in Fig. 3b show that the two peaks consistently lie on either side of the 95% ANI value.

To further test the validity of the hypothesis that the 95% ANI value can demarcate species boundaries, we examined correlation between standing nomenclature and the 95% ANI-based demarcation. As per this standard, we should expect a pair of genomes to have ANI value ≥95% if and only if both genomes are classified at the same species in the existing taxonomy. From the complete set of 89,499 genomes, we identified the subset for which we could determine the named species for each genome.

**Table 2 Accuracy evaluation of ANI values computed using FastANI and Mash**

| Dataset | FastANI | Mash | | |
|---|---|---|---|---|
| | | $-s\ 10^3$ | $-s\ 10^4$ | $-s\ 10^5$ |
| D1 | **0.995** | 0.594 | 0.932 | 0.935 |
| D2 | **0.999** | 0.996 | 0.997 | 0.997 |
| D3 | **0.995** | 0.944 | 0.944 | 0.944 |
| D4 | **0.681** | −0.040 | 0.003 | 0.010 |
| D5 | 0.998 | 0.634 | 0.997 | **0.999** |

The evaluation is done by measuring their Pearson correlation coefficients with $ANI_b$ values. Mash is executed with sketch sizes (−s): 1000 (default), 10,000, and 100,000. FastANI achieves >0.99 correlation with $ANI_b$ in all cases but D4. Its correlation value on D4 improves from 0.681 to 0.944 if the two poor assemblies present in D4 are not taken into account The best results are highlighted in bold

**Table 3 Comparison of execution time of FastANI vs. $ANI_b$**

| Dataset | FastANI | | $ANI_b$ (s) | Speedup |
|---|---|---|---|---|
| | Indexing (s) | Compute (s) | | |
| D1 | 468.2 | 16.76 | 13,113 | 782x |
| D2 | 195.7 | 264.8 | 18,155 | 69x |
| D3 | 1538 | 1981 | 99,317 | 50x |
| D4 | 128.8 | 214.5 | 11,051 | 52x |
| D5 | 2784 | 14.88 | 68,571 | 4608x |

Speedup in the last column is measured as the ratio of $ANI_b$'s runtime and FastANI's compute time

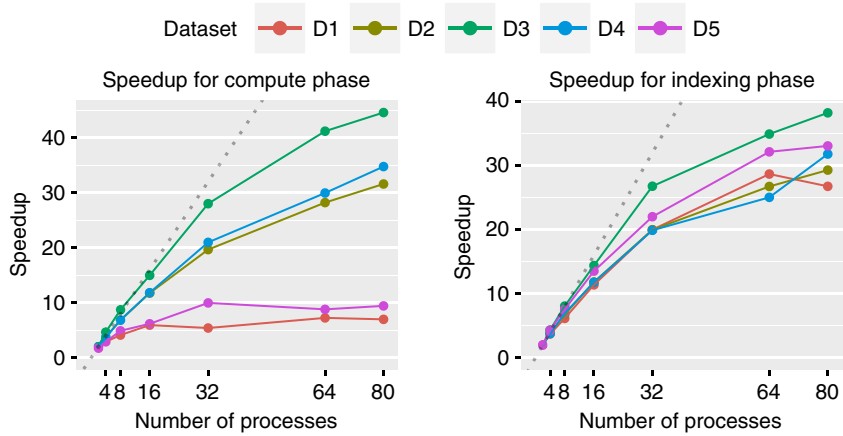

**Fig. 2** Scaling up FastANI's performance using multi-core parallel execution. We executed parallel FastANI processes on 40 physical cores, where each process was assigned an equally sized part of the reference D1–D5 databases. Left and right plots evaluate FastANI's compute and indexing phase, respectively. FastANI achieves reasonable speedups on all datasets except the compute phase in D1 and D5, as their runtime on a single core is too small to begin with (Table 3)

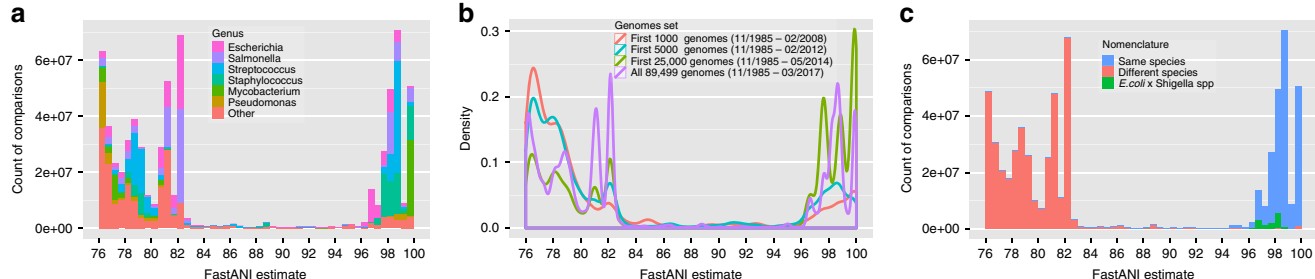

**Fig. 3** Genetic discontinuity observed using 90 K genomes. **a** Histogram plot showing the distribution of ANI values among the 90 K genomes. Only ANI values in the 76–100% range are shown. Out of total 8.01 billion pairwise genome comparisons, FastANI reported only 17M ANI values (0.21%) with ANI between 83 and 95% indicating a genetic discontinuum. Multiple colors are used to show how genomes from different genera are contributing to this distribution. Few peaks in the histogram arise from genera that have been extensively sequenced and dominate the database. **b** Density curves of ANI values in the ANI range 76–100%. Each curve shows the density curve corresponding to the database at a particular time period. Discontinuity in all four curves is observed consistently. **c**. Distribution of ANI values with each comparison labeled by the nomenclature of genomes being compared. All the comparisons between *Escherichia coli* and *Shigella* spp. have been labeled separately. The 95% ANI threshold on *x*-axis serves as a valid classifier for comparisons belonging to same and different species

Whenever available (9% of the total genomes), we recovered the assigned species using the links of NCBI taxonomy. For the remainder of the genomes, we inferred the species from the organism name given in the GenBank file, excluding all entries with ambiguous terms (sp, cf, aff, bacterium, archeon, endosymbiont), resulting in the species-wise classification of an additional 78% of the genomes. The remainder 13% of the genomes lacked clear nomenclature and hence could not be reliably assigned to a named species for the purpose of this test.

We evaluated the distribution of ANI values in comparison to the named species that the corresponding genomes were assigned to (Fig. 3c). The ≥95% ANI criterion reflects same named species with a recall frequency of 98.5% and a precision of 93.1%. We further explored the values affecting precision, i.e., 6.9% of ANI values above 95% that were obtained for genomes assigned to different named species. Among those, 5.6% are due to comparisons between *Escherichia coli* and *Shigella* spp., a case in which the inconsistency between taxonomy and genomic relatedness is well documented[1] (highlighted in green in Fig. 3c). The remaining 1.3% of the cases mostly exist within the *Mycobacterium* genus (0.5%), which includes a group of closely related named species as part of the *M. tuberculosis* complex such as *M. tuberculosis* (reference), *M. canettii* (ANI 97–99% against

reference), *M. bovis* (ANI 99.6%), *M. microti* (ANI 99.8–99.9%), and *M. africanum* (ANI 99.9%), among others. An additional 0.2% of the cases correspond to comparisons between *Neisseria gonorrhoeae* and *N. meningitidis*, two species with large representation in the database and ANI values close to 95% (Inter-quartile range: 94.9–95.2%). Excluding the cases of *E. coli* vs. *Shigella* spp. alone, precision increases to 98.7%. With both recall and precision values ≥98.5%, these results corroborate the utility of ANI for species demarcation, which is consistent with previous studies based on a much smaller datasets of genomes[4,7,15,31].

## Discussion

Our results indicate that FastANI robustly estimates ANI values between both complete and draft genomes while reducing the computing time by two to three orders of magnitude. We leveraged the computational efficiency offered by FastANI to evaluate the distribution of ANI values in a set of over 90,000 genomes, and demonstrate that genetic relatedness discontinuity can be consistently identified among these genomes around 95% ANI. This discontinuity is recovered with or without the most frequently represented species in the database (Supplementary

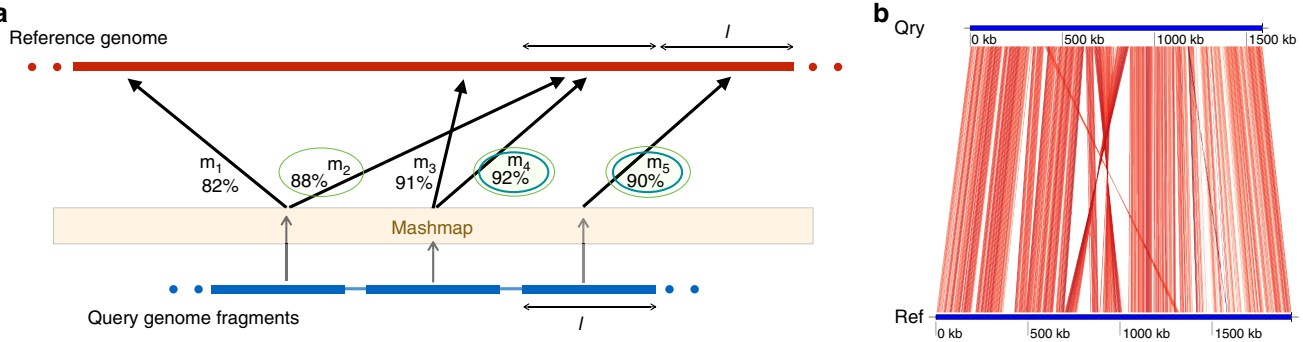

**Fig. 4** FastANI algorithm explained using synthetic and real examples. **a** Illustration of FastANI's work-flow for computing ANI between a query genome and a reference genome. Five mappings are obtained from three query fragments using Mashmap[24]. $M_{forward}$ saves the maximum identity mapping for each query fragment. In this example, $M_{forward} = \{m_2, m_4, m_5\}$. From this set, $M_{reciprocal}$ picks $m_4$ and $m_5$ as the maximum identity mapping for each reference bin. Mapping identities of orthologous mappings, thus found in $M_{reciprocal}$, are finally averaged to compute ANI. **b** FastANI supports visualization of the orthologous mappings $M_{reciprocal}$ that are used to estimate the ANI value using genoPlotR[39]. In this figure, ANI is computed between *Bartonella quintana* strain (NC_018533.1) as query and *Bartonella henselae* strain (NC_005956.1) as reference. Red line segments denote the orthologous mappings computed by FastANI for ANI estimation

Fig. 9), is robust to historic additions in the public databases (Fig. 3b), and it represents an accurate threshold for demarcating almost all currently named prokaryotic species (Fig. 3c). While this genetic discontinuity has been observed previously[4,7,15,31], the FastANI-based results reported here show a sharper discontinuity while using a much larger set of genomes by at least an order of magnitude.

The genetic discontinuity was apparent even when the species with large count of sequenced representatives such as pathogenic bacteria of human or animal hosts were iteratively removed from the analysis (Supplementary Fig. 9), or when genomes were randomly drawn with species-dependent probabilities that ensured equal representation of highly sampled and sparsely sampled species in the final set (Supplementary Fig. 10). To account for the possible influence of cultivation bias on our conclusions, we sampled five genomes from each of the 750 named species with ≥5 genomes present in the database. Even though the percentage of the inter-species pairs remains small within the 83–95% valley range (0.2%), discontinuity appears to be less pronounced (Supplementary Fig. 10). The latter was attributable to the fact that several highly sampled species have closely related species (of "intermediate" identity) that include relatively fewer sequenced representatives; thus, subsampling the genomes of the former species affected more the frequency of ANI values in the 95–100% relative to the 83–95% range. These results might indicate that cultivation biases could have accounted, at least in part, for the discontinuity observed. Cultivation biases could include, for instance, a historical tendency to preserve the isolates that meet the known/expected phenotypic criteria of the species and discard the remaining ones, which could represent "outlier" or "intermediate" genomes in terms of phenotypic and genetic similarity, or biases of the cultivation media and conditions against such "intermediate" genomes. However, given that these highly sampled species represent several distinct major prokaryotic lineages (Supplementary Fig. S8), it is likely that the discontinuity represents a real biological signature and is not driven by cultivation or other biases (or the latter should have been uniformly applied to several different isolation procedures and lineages of the highly sampled species and their close relatives). It is also important to note that these results are consistent with cultivation-independent metagenomics analysis of natural microbial communities, which have showed that the communities are composed of predominantly sequence-discrete populations[32]. Moreover, the discontinuity pattern observed using the collection of 8000 MAGs recovered from different habitats (Fig. 1, D5) is remarkably similar to the discontinuity observed among isolate genomes (Fig. 3).

The biological mechanisms underlying this genetic discontinuity are not clear but should be subject of future research for a more complete understanding of prokaryotic species. The mechanisms could involve a dramatic drop in recombination frequency around 90–95% ANI, which could account for the discontinuity if bacteria evolve sexually[33], ecological sweeps that remove diversity due to competition[34,35], or stochastic neutral processes[36,37]. A genomic nucleotide diversity of 5–10% translates to tens of thousands of years of evolution time, which provides ample opportunities for ecological or genetic sweeps to occur. Nonetheless, the existence of genetic discontinuity among 90 K genomes represents a major finding that can help define species more accurately and has important practical consequences for recognizing and communicating about prokaryotic species.

As a general-purpose research tool, we expect FastANI to be useful for analysis of both clinical and environmental microbial genomes. It can be used for studying the inter- and intra-species diversity within large collections of genomes, including genomes showing various levels of completenes (Supplementary Fig. S5). FastANI and Mash gave comparable ANI estimates for complete genomes, but the advantages of FastANI for draft (incomplete), divergent (<90% ANI) or highly related (>99.5% ANI) genomes are significant (Fig. 1), and thus, FastANI should be the preferable method. FastANI should also accelerate the study of the novelty of new species or phenotypic similarity of a query genome sequence in comparison to all available genomes.

## Methods

**The Mashmap sequence mapping algorithm**. Given a query sequence, Mashmap[24] finds all its mapping positions in the reference sequence(s) above a user specified minimum alignment identity cutoff $I_0$ with high probability. Mashmap avoids direct alignments, but instead relates alignment identity ($I$) between sequences $A$ and $B$ to Jaccard similarity of constituent k-mers ($J$) under the Poisson distribution model:

$$I(A, B)/100 = 1 + \frac{1}{k} \times \log\left(\frac{2 \cdot J(A, B)}{1 + J(A, B)}\right), \quad (1)$$

where $k$ is the k-mer size[25].

To estimate the Jaccard similarity itself, Mashmap uses a winnowed-MinHash estimator. This estimator requires only a small sample of k-mers from the query and reference sequences to be examined[24].

**FastANI extends Mashmap to compute ANI**. Previously established and widely used implementations of ANI begin by either identifying the protein coding genomic fragments[15] or extracting approximately 1 Kbp long overlapping fragments[3] from the query genome. These fragments are then mapped to the reference genome using BLASTn[10] or MUMmer[30], and the best match for each fragment is saved. This is followed by a reverse search, i.e., swapping the reference and query genomes. Mean identity of the reciprocal best matches computed through forward and reverse searches yields the ANI value. Rationale for this bi-directional approach is to bound the ANI computation to orthologous genes and discard the paralogs. In designing FastANI, we followed a similar approach while avoiding the alignment step.

FastANI first fragments the given query genome ($A$) into non-overlapping fragments of size $l$. These $l$-sized fragments are then mapped to the reference genome ($B$) using Mashmap. Mashmap first indexes the reference genome and subsequently computes mappings as well as alignment identity estimates for each query fragment, one at a time. At the end of the Mashmap run, all the query fragments $f_1, f_2 \ldots f_{\lfloor |A|/l \rfloor}$ are mapped to $B$. The results are saved in a set $M$ containing triplets of the form $\langle f, i, p \rangle$, where $f$ is the fragment id, $i$ is the identity estimate, and $p$ is the starting position where $f$ is mapped to $B$. The subset of $M$ (say $M_{\text{forward}}$) corresponding to the maximum identity mapping for each query fragment is then extracted. To further identify the reciprocal matches, each triplet $\langle f, i, p \rangle$ in $M_{\text{forward}}$ is "binned" based on its mapping position in the reference, with its value updated to $\langle f, i, bin \rangle = \langle f, i, \lfloor p/l \rfloor \rangle$. Through this step, fragments which are mapped to the same or nearby positions on the reference genome are likely to get equal bin value. Next, $M_{\text{reciprocal}}$ filters the maximum identity mapping for each bin. Finally, FastANI reports the mean identity of all the triplets in $M_{\text{reciprocal}}$ (See Fig. 4 for an example and visualization).

We define $\tau$ as an input parameter to FastANI to indicate a minimum count of reciprocal mappings for the resulting ANI value to be trusted. It is important to appropriately choose the parameters ($l$, $\tau$, and $I_0$).

**FastANI algorithm parameter settings**. FastANI is targeted to estimate ANI in the 80–100% identity range. Therefore, it calls Mashmap mapping routine with an identity cutoff $I_0 = 80\%$, which enables it to compute mappings with alignment identity close to 80% or higher.

Choosing an appropriate value of query fragment $l$ requires an evaluation of the trade-off between FastANI's computation efficiency and ANI's estimation accuracy. Higher value of $l$ implies less number of non-overlapping query fragments, thus reducing the overall runtime. However, if $l$ is much longer than the average gene length, a fragment could span more than one conserved segment, especially if the genome is highly recombinant. We empirically evaluated different values of $l$ and set it to 3 Kbp (Supplementary Table 1). Last, we set $\tau$ to 50 to avoid incorrect ANI estimation from just a few matching fragments between genomes that are too divergent (e.g., showing <80% ANI). With $l = 3$ Kbp, $\tau = 50$ implies that we require at least 150Kbp homologous genome sequence between two genomes to make a reliable ANI estimate, which is a reasonable assumption for both complete and incomplete genome assemblies based on our previous study[38].

**Code availability**. FastANI (v1.0) can be downloaded free at https://github.com/ParBLiSS/FastANI/releases.

## Data availability

All the datasets (genome sequences, accession numbers) used in this study are available at http://enve-omics.ce.gatech.edu/data/fastani.

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

## Acknowledgements

We thank Jim Cole for discussions regarding the alignment-free comparison in the FastANI algorithm. This work was supported, in part, by US National Science Foundation awards 1356288 (to K.T.K.) and CCF-1816027 (to S.A.).

## Author contributions

K.T.K. conceived the study; C.J., A.M.P., and S.A. designed method; C.J., and L.M.R. analyzed data; and C.J., L.M.R., K.T.K., and S.A. wrote the paper.

## Additional information

**Competing interests:** The authors declare no competing interests.

