## [Peer Review File · Nature Communications]

Reviewers' comments:

Reviewer #1 (Remarks to the Author):

In this paper, the authors apply an optimized method for calculating ANI (based on mashmap) to assess inter-species ANI for essentially all available bacterial and archaeal genomes. It is a combination of a method optimization paper and an observation paper. The main conclusions of the paper are that:

- (a) ANI can be calculated quickly and accurately using FastANI,
- (b) there exists an inter-species ANI "gap" between 83% ANI and 95% ANI in which essentially no pairs of bacterial/archaeal genomes exist.

This follows a fairly long thread of work by Konstantinedes et al along these lines, but is interesting and novel given the scale of the comparisons and the robustness of the result.

I have no reservations about the methods or the results after two thorough read-throughs of the paper. The citations are solid and complete, the methods are very well done, and the conclusions are well supported.

I could wish the authors used a term other than "prokaryotic" but it doesn't really matter, I suppose...

My only request for changes is that a DOI be assigned to the version of FastANI that was used in the paper, rather than a github repo (which is not archival).

Reviewer #2 (Remarks to the Author):

This paper by Jain et al. aims at answering the relevant question whether there is a continuum of genetic diversity or clear boundaries among prokaryotic species. This question is answered by considering the average nucleotide identity (ANI), which is per definition a whole-genome similarity metric, and by considering all the available prokaryotic genomes in public repositories. Since the ANI computation on a large set of genomes is computationally intensive using existing approaches, the authors propose a new method named FastANI which is able to compute ANI scores using an alignment-free approximate sequence mapping. The tool is open-source and publicly available.

The experimental analysis can be subdivided in two main parts. First, the authors show that FastANI is accurate in ANI estimation and is much faster than traditional alignment-based approaches. Then, FastANI is used to compute pairwise ANI values among all prokaryotic genomes (> 90K) available in the NCBI database. Results show a clear genetic discontinuity among the species, with 99.8% of the total genome comparisons showing either > 95% intra-species ANI or < 83% inter-species ANI values.

Overall, this is a nice contribution and the paper is well written. However, in my opinion a couple of points affect the novelty of the paper in the current form:

1. The main message of the paper from a biological point of view is that 95% ANI represents an accurate threshold for demarcating prokaryotic species. This value has been already proposed multiple times by independent studies (most notably by Konstantinidis & Tiedje, PNAS, 2005), although I acknowledge that this kind of analysis is conducted in the current paper on a significant larger number of samples. However, it is more a confirmation than a new finding.

2. The main competitor of FastANI is represented by MASH. Although comparisons among the two methods are reported in some parts of the manuscript (see Figure 1 and Table 2), the paper lacks their comparison on the 90K genomes. I would expect to see two main comparisons in this direction:
i) FastANI took around 77K CPU hours for computing pairwise ANI values for the entire database. Which is the computational time for MASH on the same set of samples?;
ii) could you report the same three plots shown in Figure 3 for FastANI also for MASH? Is the 95% ANI threshold evident from the MASH results too?

Other comments:

3. Comparisons among FastANI and MASH (see Table 1-2, Figure 1) are done by taking into account five different datasets, but by considering only one species as query genome per dataset. Could you increase the number of query genomes per dataset?

4. I am bit surprised that MASH outperformed FastANI for the dataset D5 (see Table 2). From my understanding this should be the most unfavorable dataset for MASH since it includes a large amount of non-complete genomes. Could you comment on this?

5. I thank the authors for making the code freely available. As written by the authors on the website, "FastANI doesn't support parallelization internally" (something that for example is implemented in MASH). Many-to-many genome comparisons rely on an external script, which is however a bit tricky to use. Would it be possible to implement a real parallelization? I think this is something really important to make the tool really scalable to large set of genomes.

Reviewer #3 (Remarks to the Author):

The manuscript by Jain et al. "FastANI: High-throughput ANI Analysis of 90K Prokaryotic Genomes Reveals Clear Species Boundaries" describes a scalable method of alignment-free estimation of genome-wide Average Nucleotide Identity (ANI), which is one of the most popular and accurate methods for assignment of genomes to species-level groups. The manuscript consists of two parts: characterization of FastANI method itself, and its comparison to BLAST-based ANI computation, as well as to the fast, but less accurate alignment-free method of ANI estimation, Mash. This benchmarking is done using 5 test datasets of varying quality and diversity, ranging from closed genomes to a large collection of metagenome-assembled genomes (MAGs). The authors demonstrate that, overall, FastANI delivers a speedup of several orders of magnitude as compared to alignment-based methods, and also provides a better ANI estimate than Mash, achieving >0.99 correlation with BLAST-based ANI values on most datasets.

The authors also reproduce the results of some previous ANI studies, which showed that ANI-based species demarcation recapitulates existing taxonomic assignments with high accuracy.

The second part of the manuscript describes an application of FastANI method to the largest to date collection of prokaryotic genomes in order to assess whether a species demarcation cutoff of ~95% ANI, which was proposed in earlier smaller studies, still holds as the number of genomes continues to grow. The authors show that pairwise ANI estimates computed by FastANI follow a bimodal

distribution, with the vast majority of genome pairs having less than 83% or more than 95% ANI. They also demonstrate that bimodal distribution is robust to various data subsampling strategies, including by the time of data deposition, or by species representation. The authors conclude that bimodal distribution of pairwise ANI estimates is a real biological phenomenon, and not an artifact of sampling, sequencing, or cultivation.

Overall, while the findings aren't very novel, the manuscript reports on a very useful method with many applications, and may be of interest to the wide community of microbiologists, microbial ecologists, computational biologists, etc.

I have three serious issues with the manuscript:

1. the authors are absolutely right to suggest that the ability to quickly identify genomes of the same species is crucial given the amount of sequencing data being generated. Unfortunately, in the past year or two it is mostly generated in the form of MAGs and SAGs, which are inherently incomplete, much more prone to misassemblies than traditional isolate genomes, and also suffer from contamination from various sources. Yet instead of performing a proper benchmarking study using simulated data with known degree of completeness, misassembly and contamination, the authors use an estimate of completeness based on CheckM, and then report what amounts to anecdotal evidence for misassembly and contamination.

While CheckM is widely used, it is far from infallible. Furthermore, genomes with varying completeness, misassembly and contamination are not that difficult to simulate. Similarly, the cutoff for the number of segments with pairwise ANI hits may need to be revisited given the quality of the data to which this tool will likely be applied. In addition, the data shown in Fig. S4 is quite puzzling, and do not conform with what contaminated genomes normally look like; in the absence of strain identifiers and accessions readers cannot find these genomes in the databases and evaluate their quality using other tools.

2. no accessions are provided for genome sequences included benchmarking datasets, which makes it difficult to locate the specific genomes discussed by the authors.

3. the authors use rather confusing terminology to describe their observations and results. My main gripe is with the use of the term "discontinuity" (also appearing as "discontinuum") with a variety of adjectives, such as "wide". Having "wide discontinuity" is somewhat akin to being "slightly pregnant", and should be replaced with a better term describing a bimodal distribution of ANI estimate. This terminology is used throughout the manuscript, and there are too many instances to list each one separately.

Reviewer 1

1. *Assign a DOI to the version of FastANI that was used in the paper.*

We have specified the version number of FastANI in the revised manuscript. This version is easily accessible as a separate release on the Github page.

Reviewer 2

1. *What is the computational time for MASH to process the set of 90K genomes? Is the 95% ANI threshold evident from the MASH results too?*

Mash results on the set of 90K genomes have been added in Supplementary Fig. S7. Mash produces a bimodal distribution similar to FastANI. The 95% ANI threshold is evident from Mash results as well. In the distribution obtained using Mash, the shape of the first peak appears different from FastANI (Figure 3c). Distinct shape of left peak appears because of its less accuracy on moderately divergent genomes (i.e., showing 80-90% ANI).

Mash implements a much simpler logic, therefore it is orders of magnitude faster than FastANI, and speedups vary depending on the sketch size used (Supplementary Fig. S7). This speedup however, comes at the cost of low accuracy because Mash does not compute the shared genes of genome pairs (Fig. 1).

2. *Comparisons among FastANI and Mash (see Table 1-2, Figure 1) are done by taking into account five different datasets, but by considering only one species as query genome per dataset. Could you increase the number of query genomes per dataset?*

We have added results of the experiment done with five randomly picked query genomes in Supplementary Fig. S3. Conclusions which were originally made from Table 1-2 and Figure 1 continue to hold with new results as well. Note that we cannot scale this particular experiment to a large set of query genomes because BLAST-based ANI solver is too slow.

3. *Mash outperformed FastANI for the dataset D5 (see Table 2). From my understanding this should be the most unfavorable dataset for MASH since it includes a large amount of non-complete genomes. Could you comment on this?*

Mash with sketch size 10^5 produces marginally better results than FastANI using dataset D5 (Table 2). Genome completeness (ranging from 75%-100%, Supplementary Fig. S1) alone may not affect Mash results much as long as the shared fraction of two genomes is sufficiently high, and contamination/mis-assembly is kept low.

4. *Would it be possible to implement a real parallelization? I think this is something really important to make the tool really scalable to large set of genomes.*

Yes, we have released a new version of FastANI which implements multi-threading.

Reviewer 3

1. *Instead of performing a proper benchmarking study using simulated data with known degree of completeness, misassembly and contamination, the authors use an estimate of completeness based on CheckM. While CheckM is widely used, it is far from infallible.*

We have added an experiment which evaluates FastANI using various levels of completeness and contamination levels using simulation (Supplementary Fig. S5). FastANI continued to show good resilience to high levels to contamination and incompleteness in the simulated input genomes.

2. *The data shown in Fig. S4 is quite puzzling, and do not conform with what contaminated genomes normally look like; in the absence of strain identifiers and accessions readers cannot find these genomes in the databases and evaluate their quality using other tools.*

Strain identifiers in Fig. S4 have been added.

3. *No accessions are provided for genome sequences included benchmarking datasets, which makes it difficult to locate the specific genomes discussed by the authors.*

We have uploaded the accession numbers of all the genomes on our website. The “Data Availability” section has been revised accordingly.

4. *Misuse of the term “discontinuity” with a variety of adjectives.*

We have corrected this in the revised manuscript.

REVIEWERS' COMMENTS:

Reviewer #2 (Remarks to the Author):

The authors answered most of my comments and it is a very good news that they implemented a parallel version. The authors did not comment on the fact that there is limited novelty in observing that the 95% ANI threshold is good for prokaryotic species demarcation. Overall, I think the paper is in a better shape. I feel that, overall, Mash remains a challenging tool for FastANI to overcome, as the improved speed of Mash can justify a minor loss in accuracy, and the 95% identity species demarcation is highlighted also by Mash. I think the authors should better highlight this in the paper.

Reviewer 2

1. *The authors did not comment on the fact that there is limited novelty in observing that the 95% ANI threshold is good for prokaryotic species demarcation.*

Yes, 95% threshold has been observed previously, and these papers have been appropriately cited during the introduction and results sections. Existing studies either used small number of genomes, or they presented results based on universal marker genes which led to inconsistent results (the marker genes of the genome evolve slower than the genome average). No study was done at this scale previously, and the gap is more sharp using more data compared to the earlier efforts. We have updated the Discussion section to better highlight this.

2. *I feel that, overall, Mash remains a challenging tool for FastANI to overcome, as the improved speed of Mash can justify a minor loss in accuracy, and the 95% identity species demarcation is highlighted also by Mash.*

Our primary motivation for FastANI is to maintain high ANI accuracy for the input genome sequences that are incomplete or draft-quality, which typically happens for MAGs and SAGs. Using simulations (Supplementary Figure 5), we show that FastANI can tolerate high level of contamination and incompleteness in the genome sequences. FastANI and Mash gave comparable ANI estimates for complete genomes, but the advantages of FastANI for draft (incomplete) or divergent (<90% ANI) or highly related (>99.5% ANI) genomes are significant (Figure 1), and thus, FastANI should be the preferable method, even though it is slower than Mash, in general. We have revised the Discussion section accordingly to highlight this advantage.